# Analysis of Correlation Effects of Double Mutations in Enzymes: A Revised Residual-Contact Network Clique Model

**DOI:** 10.3390/ijms25169114

**Published:** 2024-08-22

**Authors:** Xianbo Zhang, Junpeng Xu, Dengming Ming

**Affiliations:** College of Biotechnology and Pharmaceutical Engineering, Nanjing Tech University, 30 South Puzhu Road, Jiangbei New District, Nanjing 211816, China; 202121019056@njtech.edu.cn (X.Z.); 201761101184@njtech.edu.cn (J.X.)

**Keywords:** double mutation, correlation effect, additivity, residual-contact network, clique community

## Abstract

The relationship between amino acid mutations and enzyme bioactivity is a significant challenge in modern bio-industrial applications. Despite many successful designs relying on complex correlations among mutations at different enzyme sites, the underlying mechanisms of these correlations still need to be explored. In this study, we introduced a revised version of the residual-contact network clique model to investigate the additive effect of double mutations based on the mutation occurrence topology, secondary structures, and physicochemical properties. The model was applied to a set of 182 double mutations reported in three extensively studied enzymes, and it successfully identified over 90% of additive double mutations and a majority of non-additive double mutations. The calculations revealed that the mutation additivity depends intensely on the studied mutation sites’ topology and physicochemical properties. For example, double mutations on irregular secondary structure regions tend to be non-additive. Our method provides valuable tools for facilitating enzyme design and optimization. The code and relevant data are available at Github.

## 1. Introduction

Understanding the impact of changes in a protein’s amino acid sequence on its three-dimensional structure and biological activity plays a decisive role in the success of the rational design of engineered enzymes [1]. For this reason, point mutation remains the primary method for studying enzyme function and the most commonly used tool for rational modification of engineered enzymes. If multiple site mutations do not interfere with each other, then in theory, significant performance improvements can be achieved by simply superimposing multiple single-point mutations with similar functions [2]. For example, combining a series of site mutations that slightly increase stability can significantly improve the overall stability of a protein. However, the reality is that multiple site mutations often interfere with each other, and their contributions to protein performance are likely to offset or even deteriorate each other [3]. The final overall effect is random, deviating from the simple superposition effect. Therefore, it is essential to accurately predict the additivity relationship between mutations at different sites and characterize the degree of correlation between amino acid mutations. It can filter out malicious-related mutations in advance, significantly reducing the experimental workload and improving design efficiency.

Structural biologists have carried out studies on mutational additivity in different contexts. Wells and colleagues [4] in 1990 demonstrated that combinations of mutations affecting protein–protein interactions, DNA–protein recognition, or protein stability show simple additivity in most cases. Still, there is a significant bias in the additivity of mutations when mutation sites interact strongly with each other (e.g., by direct contact, indirect forces, or spatial perturbations). Interestingly, the opposite condition also exists. In addition, they found that the additivity of sites involved in van der Waals interactions was weakened. Boyer and colleagues believed that the non-additivity of distal sites indicated the existence of information communication between amino acid sites, and they called this thermally stable non-additivity phenomenon thermodynamic coupling [5]. They tried to determine the range of perturbations between two mutation sites, concluding that amino acid interactions in the natural structure of the protein caused thermodynamic coupling of distal sites. In 1992, Sondek and colleagues studied the structural and energetic differences caused by multiple mutations in adjacent amino acids in staphylococcal nuclease (SNase) [6]. SNase is an extracellular enzyme produced by Staphylococcus aureus and is composed of 149 amino acids. The following year, Green and colleagues constructed 71 double mutants and analyzed the interaction between stability and amino acid position in SNase. They found that sequentially distant but structurally close mutants exhibit significant non-additivity [7]. Later, Chen and colleagues constructed 44 double mutants at the hydrophobic core of SNase and systematically investigated the additivity of these mutations by comparing them with the corresponding single mutations [8]. In 1993, Terwilliger and colleagues examined double mutants of the phage F1 gene V protein and found that both protein stability and DNA binding affinity were approximately equal to the sum of the effects of the two individual mutations [9]. In 1996, they investigated double mutations in the hydrophobic core region of the phage F1 gene V protein and suggested that the additivity of two mutations could be determined by examining whether there was an overlap in the regions affected by single mutations [10]. In a series of studies on T4 phage lysozyme by Matthews and colleagues [11], the additivity and non-additivity of mutations under different mutation combinations are discussed. Notably, Matthews and colleagues argued that dynamic perturbations from mutations would start at the mutation site and spread to its neighbors. A series of different changes at sites close to the mutation sites would lead to very different results in terms of additivity and non-additivity.

Recently, Jemimah et al. [12] investigated the impact of the additivity of double mutants in protein–protein complexes on protein binding affinity. They compared the binding free energies (∆∆G-bind) of 379 double mutants with the algebraic sum of the binding free energies of the corresponding single mutants. They found that double mutants with non-additive effects tend to have the mutated amino acids closer together and may even have some degree of contact. Ming and colleagues [13] proposed a topology-based model to model the non-additivity effect in T4 phage lysozyme mutations [14,15,16]. An amino acid interaction network was established on the conformations of a series of mutant proteins obtained from molecular dynamics simulations (MD), based on which a 3-clique community was constructed to identify the amino acid sites that affect each other, to make a judgment on the additivity or non-additivity of the double mutations. However, systematic research on the correlation effects of multi-mutations is still lacking, and many studies have been limited to individual cases [17,18].

This paper presents a modified residual-contact network clique model to investigate the additivity effects of double mutants in staphylococcal nuclease from *S. aureus* and the gene V protein from phage F1. The model was an improved version of our original protein contact map model originally applied to the phage T4–lysozyme mutation correlation. It constructed a series of double mutant structures using the newly released structure-building program AlphaFold2 [19] to consider structural changes brought about by mutations. The improved model identifies the additivity of the sites in the mutations with higher accuracy and lower computational power, corrects the previous biased prediction results, and identifies some exceptional cases in the additivity. Still, with the construction of a specific double mutant protein structure by AlphaFold, we carried out a particular study on examples of model bias and found the reasons for model bias and the special effects of mutations.

## 2. Results and Discussion

### 2.1. Additivity Analysis of Double Mutations in Staphylococcal Nuclease

We counted the double mutation stability data of 119 pairs of Staphylococcal nuclease (Appendix A), and Figure 1 shows that most of the amino acid double mutations in staphylococcal nuclease have an additivity effect. However, some of the mutant structures have a significant deviation from the cumulative value of single mutations, indicating that there is non-additivity.

#### 2.1.1. Clique Analysis of Double Mutation Correlation in Staphylococcal Nuclease

Figure 2 shows the 3-Clique communities of staphylococcal nuclease based on native structure analysis; detailed results are provided in Appendix A. Larger 3-Clique Communities tend to form in the α-helix, while multiple close but unconnected 3-Clique Communities tend to occur in the sheet area.

After examining the existence of double mutant pairs in the 3-clique community (Appendix A), it was found that double mutant pairs in the wild-type structure with both sites existing in the 3-clique community were present in only six pairs of the selected data (L37/A90, T62/V66, I92/V99, G79/N118, P117/N118), unable to make compelling predictions for datasets. However, different residues introduced by the mutation all had different effects on the structure and function of the protein; this means that additive or non-additive prediction of a large number of double mutations based on the wild-type structure or a small number of known protein structures will become problematic. To further explore how the changes in residues brought about by mutations affect the thermodynamic stability of proteins, we predicted the structure of each double mutant protein using AlphaFold, recalculated the 3-clique community distribution in the mutated proteins, and examined the relationship between mutational additivity and 3-clique community (Appendix A). An alignment calculation showed that the structures of the mutant AlphaFold models are very close to that of the wild-type protein, with a very small RMSD value of 0.88 angstroms.

In 119 pairs of double mutation data collected, ten mutations showed non-additivity and 109 pairs showed additivity. The prediction of mutational additivity by the model showed that there was no 3-clique community between 106 pairs of double mutations. According to the model prediction, these combinations should show mutation additivity. Still, by comparing it with the experimental data, it was found that 7 out of 106 pairs of mutations were non-additive, which is not the case in the actual situation. Among the remaining 13 pairs of mutation combinations that formed a 3-clique community, 10 pairs of data exhibited mutation additivity but failed to be correctly predicted by the model. The model’s accuracy in identifying double mutation additivity is 93.40%, while the accuracy for non-additivity is only 23.08%; for double mutation data in the entire dataset, the model successfully predicted 99 of the 109 additive double mutations with 90.82% accuracy. For the ten pairs of non-additive data, the model predicted only three pairs of sites with an accuracy of only 30.00%.

#### 2.1.2. Perturbations in Clique Due to Local Structural Changes in Double Mutations

In the 3-clique community results of the recalculations, the double mutant pairs of E75V/A90S, V23F/T33S, F61A/T62A, and L25A/M26A formed a new 3-clique community in the structure obtained by AlphaFold calculations. Through examination, it was found that in most of the double mutants that did not create the same 3-clique community with each other, $\left|∆∆∆G\right|\;\leq \;1$ kcal/mol showed the additivity of the mutation. Meanwhile, G79S/P117L vanished from the originally existing 3-clique community in the new structure, which is fulfilled with the model’s definition of additivity, i.e., when the double mutation sites did not co-exist with a 3-clique community, the double mutation tended to appear additive.

#### 2.1.3. Hydrophobic Core and 2nd Structure Fragment in Additive Double Mutations in Clique Structures

Of ten pairs of double mutations with $\left|∆∆∆G\right|\;\leq \;1$ kcal/mol (Table 1), two mutation sites form the same 3-clique community with each other and also exhibit mutational additivity. In the 1–8 indexed mutation pairs, the new clusters introduced more *hydrophobic amino acids*, which made the protein tend to have stable and ordered water structures around it, contributing to a more stable protein folding state of the double mutation compared to the single-point mutation, and presented a stronger mutational additivity. For example, in the L37A/A90S mutation pair, the 3-clique community changes from wild-type 90A/91Y/37L to 99V/100N/37A/36L/90S/92I, and the hydrophobic amino acids in the community increase from two to four. In addition, E75V/A90S and V23F/T33S showed weak mutational additivity due to the low number of hydrophobic sites in the new cluster formed.

Regarding proteins’ secondary structures, these communities are formed mainly in the same or close secondary structure (Figure 3). The large number of hydrogen bonds formed during protein folding can make regular secondary structures such as α-helix and β-sheet more stable, and larger communities indicate stronger interaction forces, including hydrogen bonds. At the same time, forming a more stable secondary structure contributes to gathering hydrophobic amino acids within the protein and creating a hydrophobic core, which significantly reduces the free energy of the protein during the folding process.

#### 2.1.4. Non-Additive Double Mutations in Clique Structures

Table 2 shows the case of staphylococcal nuclease double mutant $\left|∆∆∆G\right|>1$ kcal/mol, in which three pairs of double mutation sites, G79S/N118D, G79D/N118D, and P117L/N118D, formed a 3-clique community. There were strong interacting forces between the mutations, which showed *non-additivity* in the additivity of the mutations. The rest of the mutation pairs that did not form a 3-clique community with each other directly also showed strong non-additivity. This non-additivity may be indirectly affected by different amino acid sites.

#### 2.1.5. Outside the Clique: Effect of the Third Amino Acid Site between Double Mutations

David and colleagues, in their early studies on staphylococcal nuclease mutants, had reported a mutation shielding effect between the three sites 37/79/118; with the impact of the mutation at site 79/118 dramatically disappearing after it was added to the mutant at site 37 [7]. We examined the protein’s primary structure and found that the shielding effect between the three sites, 37/79/118, is strongly related to site 77. The L37A single mutant has no interaction force between the 37A and 77D sites (Figure 4A). In contrast, in the two single mutants of G79S (Figure 4B) and N118D (Figure 4C), van der Waals force interaction occurs between the 37L and 77D sites. There is always a stable interaction between the 77D and 118 sites van der Waals forces, and there has been stable hydrogen bonding between sites 79 and 118. After the mutation was introduced simultaneously between the two sites 37/79 (Figure 4D) and 37/118 (Figure 4E), the van der Waals forces that should have existed between 37 and 77 due to the effect of the mutations in 79 and 118 disappeared. Indeed, the stable system formed by the mutations at sites 79 and 118 was disrupted by the introduction of the mutation at site 37, making the new double mutant system closer to the 37A single mutant structure in terms of how the forces work on the primary structure, and this can be reflected in the ∆∆G of the double mutants, which is much closer in energy to that of the single L37A mutant for the L37A/G79S and L37A/N118D double mutants.

A similar phenomenon was observed in the double mutants L37A/E75V and V23F/I72V. There was no direct interaction between the two mutation sites, but rather an indirect effect through a third or even multiple amino acid sites, leading to mutation non-additivity (Appendix A).

It is confusing that neither 7L nor 7A in single or multiple mutants interacts directly with other amino acids, suggesting that for L7A-containing double mutants, the two mutations should be independent and possess a certain kind of additivity. Still, instead, in the two mutants, L7A/L37A and L7A/G79S, a strong non-additivity is shown. A comparison of the overall force patterns and 3-clique community of these two double mutants and the single mutant indicates that, despite the absence of direct interaction forces, the introduction of L7A appears to have affected the force patterns of some areas of the protein in a particular way.

Our studies on the non-additive site of staphylococcal nuclease have shown that the mentioned mutations (7/37/72/75/79/118) are located in irregularly coiled and turning structures. Both of them are usually not involved in the formation of the secondary structure of proteins, and the mutations may change the flexibility or polarity of these areas, impacting the kinetic properties of the protein and the overall stability. In addition, the irregular region plays a vital role in the protein folding process, and different mutations may lead to different folding pathways of the protein, which, as a result, affects the final free energy of folding. These features of irregular regions and turns in double mutants may lead to a nonlinear relationship between free energies, which is ultimately reflected in the non-additivity of the mutations.

### 2.2. Double Mutations in Gene V

Figure 5 shows the mutational additivity in these double mutants, with 29 pairs of mutation combinations showing a strong linear relationship, suggesting that they possess a certain degree of mutational additivity. Three pairs of double mutants (C33M/I47C, L65P/F68L, V35A/I47A) showed non-additivity based on thermodynamic stability.

#### 2.2.1. The 3-Clique Analysis of Gene V Protein

The 3-clique community results for the wild-type of the phage F1 gene V protein are shown in Figure 6; detailed calculation results are in Appendix A. Since only a short α-helix, consisting of sites 65–68, is present in the structure of the gene V protein, it does not form a large and tightly closed 3-clique community like the staphylococcal nuclease. After predicting the protein structure introducing the mutation by AlphaFold, we recalculated the 3-clique for the mutant to obtain the results (Appendix A).

In the additivity prediction for Gene V, 31 pairs of mutations that did not form a 3-clique community were considered additive. After checking the experimental data, only two sets of double mutations showed non-additivity, and the model predicted additivity with 93.5% accuracy. In the dataset, only one set of double mutations, C33M/I47C, formed the 3-clique community and was considered a non-additive site, which was verified to be consistent with the experimental data. The model successfully predicted all of the 29 additive double mutations. The model predicted only one pair of sites with an accuracy of 33.3% for the three non-additive sites.

#### 2.2.2. Clique Perturbation Due to Local Structural Change by Mutations

One interesting observation in this protein is that the clique community of the double mutant F13T/E30F disappeared after mutations at sites 13 and 30, which were initially present in the wild-type and showed additive results. This is consistent with our assumption that mutations that are not in the same clique tend to be additive. F13T/E30F is an example of a mutation-induced local structural change that alters the clique community to a certain extent. The new results are more helpful in explaining some of the previous unusual results of mutational additivity.

For the three pairs of combinations with |∆∆∆G|>1 kcal/mol, the model showed some bias. C33M/I47C formed a 3-clique community with 81, 33, and 47 sites after the mutation, and an interaction force was formed between the sites, which led to the non-additivity of the mutation. In contrast, the L65P/F68L mutation sites did not form a 3-clique community with each other (Figure 7). After examination, it was found that a hydrogen bond was formed directly between L65P and F68L, which contributed to the structural bending of the β-sheet. Still, since the two sites only interacted, a third site of action was absent, failing to form a clique with the three amino acid minimum requirement, much less a 3-clique community.

#### 2.2.3. Prediction Bias Due to Different Bases for Judging Mutation Correlations

Despite the occurrence of |△△△G|>1 kcal/mol, the V35A/I47A double mutant did show additivity between the two sites in terms of the folding free energies of the single mutants V35A (−2.3 kcal/mol) and I47A (−7.1 kcal/mol). A significant bias in judging the additivity at a threshold of 1 kcal/mol was observed due to the large change in the free energy brought about by the I47A mutation. When this case was studied on the judgment basis proposed by Andrei Y. Istomin et al. in their study of rigid clusters [20], the value obtained was 0.149, less than the threshold criterion of 0.2, confirming the existence of inter-site additivity.

### 2.3. Improvement to the Model: Double Mutations in Phage T4 Lysozyme

In a study of mutational additivity in staphylococcal nuclease, we found that the 3-clique community formed between two mutations will present mutational additivity if it is present in the same or two different but tightly interacting secondary structures of the protein. This effect is pronounced in the α-helix, where the introduction of mutations will form a larger, more tightly ordered hydrophobic core, allowing the protein thermodynamics to have better stability. For this reason, we introduced a new criterion for model prediction of additivity: if a pair of double mutations with a 3-clique community is located in a single α-helix, then it should be additive, and if it is located in different but tightly interacting α-helixes, it can be used as a potentially additive combinatorial site.

The protein structure of phage T4 lysozyme contains a large number of α-helices (Appendix A), and we collected 31 pairs of data on the thermodynamic stability of double mutations in phage T4 lysozymes to determine how well the model identifies thermodynamic mutation additivity after the introduction of the new criterion (Appendix A).

The calculations showed that a total of 13 pairs of double mutations formed a 3-clique community, with 8 pairs of mutations located in the same α-helix (Table 3 index 1–8) (Appendix A) and 5 pairs of mutations located in neighboring tightly interacting α-helixes (Table 3 index 9–13) (Appendix A). We considered these eight pairs of double mutations in the same α-helix as additive mutation combinations and the five pairs of double mutations as potentially additive combinations. After comparing the experimental results, all eight groups of double mutations within the same α-helix were additive, and four of the five groups of potentially additive double mutations showed additivity. The results demonstrate that with the revised clique model, the model identifies previously ignored or misidentified double mutation combinations and identifies potential double mutation combinations as a new reference for follow-up experiments.

## 3. Methods

### 3.1. Protein Mutation Datasets

We screened the protein thermodynamics data in the ProThermDB database [21]. We selected a series of single and double mutations containing data on $∆∆G_{exp}$ kcal/mol (the free energy of protein unfolding obtained under the thermal denaturation method) and $∆∆G_{H2O}$ kcal/mol (the free energy of protein deconvolution in water). Double mutations were examined to see if they also had summed data for the two single mutations, and cases that matched were retained for further analysis. The selected single and double mutation data were as consistent as possible under experimental conditions. In total, we collected double mutation data for 119 pairs of staphylococcal nuclease samples, 32 pairs of gene V samples, and 31 pairs of T4 phage lysozyme samples.

### 3.2. Determining the Additivity and Non-Additivity of Double Mutations 

For the obtained data, we studied the additivity of mutations using the following equation:(1)$$∆∆G_{sum}=∆∆G_{i}+∆∆G_{j}\;∆∆∆G=∆∆G_{exp}-∆∆G_{sum}$$
where $∆∆G_{sum\;}$ is obtained by summing the data $∆∆G_{i}$ and $∆∆G_{j}\;$of two corresponding individual mutations, $∆∆G_{i}$ and $∆∆G_{i}$ are the Gibbs free energy changes due to the mutation of amino acid sites i and j, $∆∆G_{exp}$ is the change in Gibbs free energy of the double mutant containing i and j, and ∆∆∆G indicates the difference in thermodynamic effects of the double mutant and two single mutants. The larger absolute value of ∆∆∆G indicates that the two mutation sites tend to be more non-additive. On the contrary, the smaller absolute value of ∆∆∆G suggests that the two mutations are more likely to be additive to the mutation.

Andrei and colleagues studied the rigid cluster model by considering double mutation combinations with $\delta (i,j)=|(\Delta \Delta G_{exp}-\Delta \Delta G_{sum})/\Delta \Delta G_{exp}|>0.2\;$as non-additivity sites [20]. Data analysis indicated that under this judgment, the model prefers to consider that non-additivity exists in general between internal protein sites (Figure 8). Meanwhile, the threshold δ(i,j) has a significant error at small values of ΔΔ*G_exp_*. Therefore, in this study, we adopted $\Delta \Delta \Delta G=|\Delta \Delta G_{exp}-\Delta \Delta G_{sum}|>1.0\;$kcal/mol as a standard for determining whether a mutation combination is additive or not.

### 3.3. AlphaFold Modeling of the Specific Structure of Double Mutations

Based on previous work, we considered that the different amino acids introduced by mutations would affect the computational results of the clique community in various ways. This may be due to the fact that the mutations introduce some degree of local structural alteration. To this end, we used the program AlphaFold2 [22], developed by DeepMind, to build structural models for a series of mutants and to study the additivity and non-additivity of mutations. AlphaFold builds accurate 3D structures of proteins by predicting the distribution of distances between each pair of amino acids in a protein, as well as the angles between the chemical bonds connecting the pairs of amino acids. It summarizes the measurements of all pairs of amino acids into a histogram of distances and learns from these mutations through its innovative attention-based neural network. By deploying ColabFold on a local server [23], fast and large-scale protein structure prediction can be realized. Usually, 5 predicted structures are given by default, and the highest-ranked structure is selected as the mutated structure after sorting according to the pLDDT value.

### 3.4. Residual-Contact Network

Amino acid networks (AANs), based on protein 3D structures, offer new ways to study protein structure and function. It has been applied in many studies, including identification of functional protein residues, prediction of protein folding, and protein stability analysis. As a type of AAN, the residue interaction networks (RINs) represent the contacts of amino acid residues in protein structures. We used the Residue Interaction Network Generator (RING-4.0) to calculate non-covalent interactions such as hydrogen bonds, van der Waals forces, electrostatic interactions, etc., and based on this, we built AANs for follow-up studies [24].

### 3.5. The Clique Community and the Correlation Effects in Double-Site Mutations

Specific amino acid graph networks could be constructed using the Networkx package version 2.4 [25] and studied by considering amino acid sites as nodes in the amino acid network and interaction forces between amino acids as edges in the graph. The triangle-like pattern of forces between multiple amino acids is defined as a 3-clique community, and amino acid sites within the same 3-clique community are considered to be strongly correlated with each other and exhibit non-additivity in mutations. In previous work, Ming and colleagues examined the double mutation site of the T4 phage lysozyme in the 3-clique community and made predictions about non-additivity between the sites. $P_{ab}$ determines whether a pair of mutation sites are additive or non-additive by calculating the frequency of formation of a 3-clique community between the two sites in a series of conformations obtained from a molecular dynamics simulation. This method makes a uniform calculation of known data in several homology structures that have been constructed without considering the effects of specific mutations. There may be bias in the judgment of non-additivity between some sites.

Meanwhile, the molecular dynamics simulation consumes a tremendous amount of arithmetic power and time, which is not conducive to prediction when large-scale mutation experiments are carried out. Therefore, we constructed the specific structure of double mutants using the program AlphaFold 2. Given the mutant structure, two mutations are predicted to be non-additive if they are located in the same 3-clique community; conversely, they are considered to be additive if they do not form a 3-clique community. To account for more complicated scenarios that cannot be covered by this simple rule, we have added the following additional terms to the model: (1) When two mutations are introduced into the same α-helix to form a new, larger hydrophobic core, the model predicts that it is an additive double mutation even when a 3-clique community is formed; (2) Two mutations in adjacent α-helices that form a 3-clique community typically show additive effects. These mutation pairs may be considered potential additive sites; (3) Two interacting mutations that are isolated from others cannot form a 3-clique community, yet they may still exhibit non-additivity. Calculations were performed using Python coding language (see Github: https://github.com/mingdengming/rcnc (accessed on 20 July 2024)). Under the assumption of case-specific analysis, the 3-clique community analysis was implemented directly on the mutated structures, and a large number of predictions of double mutant structures was obtained with high efficiency.

## 4. Conclusions

The study of protein mutations, particularly the design and analysis of multi-mutations, represents a frontier in biotechnology with immense potential for impact across various fields. The challenges inherent in this endeavor—from the vast combinatorial space of possible mutations to the complex, often non-linear interactions—necessitate continued innovation in experimental techniques and computational approaches [26,27]. To this end, this paper presented a revised residual-contact network model to explore the relatively fundamental problem of correlation effects between two mutations. The model extends our recently developed clique model. It introduces the characterization of the local mutation structure, the physicochemical features of mutant amino acids, and the secondary structure inside the clique. We collected 182 double mutation data reported in three extensively studied enzymes, including the staphylococcal nuclease and the gene V protein of Phage F1 and Phage T4 lysozymes, to validate the model. The model successfully identified more than 90% additive double mutations and a majority of non-additive double mutations.

We also noticed that challenges in accurately modeling non-additive double mutations remained. Mutations in different clique communities or outside of the clique may have non-additive interactions through a third amino acid and deserve further modeling studies. For example, in staphylococcal nuclease, sites of non-additive mutations are found in irregularly coiled structures, and residues in these regions usually have less contact with surrounding amino acids. As a result, they often fail to form clique structures and, according to our model, do not belong in the non-additivity category. In the case of gene V protein, there are double mutants where only two residues in random coils are in contact without a third being involved, but they are non-additive.

Taken together, our calculations show that most additivity and partial non-additivity effects in double mutations can be deciphered by studying the mutation sites’ topological structure and physicochemical properties of constituent amino acids. At the same time, we may need to consider certain intrinsic indirect interactions between residues not included in the residual-contact map to better model the non-binding effects of double mutations.

## Figures and Tables

**Figure 1 ijms-25-09114-f001:**
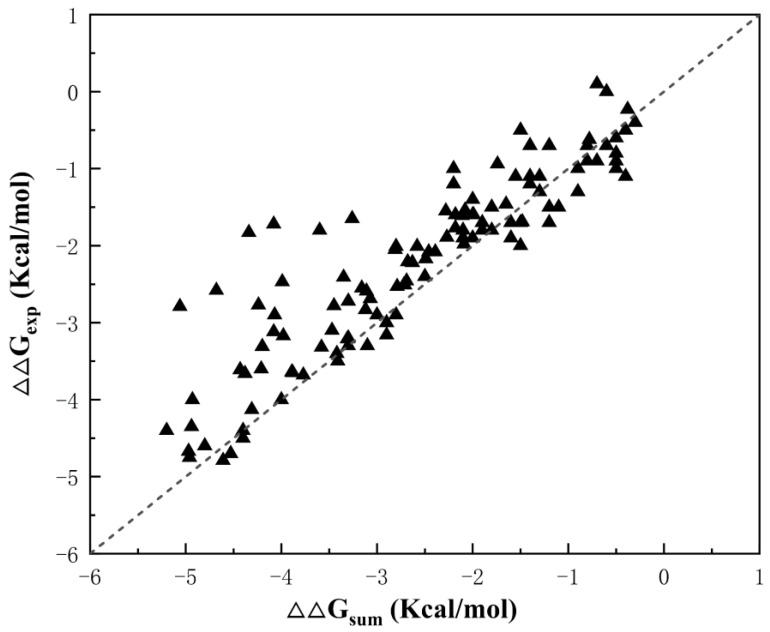
Distribution of thermodynamic data for staphylococcal nuclease double mutations, where $∆∆G_{sum}=∆∆G_{i}+∆∆G_{j}$ and $∆∆G_{exp}$ is the unfolding free energy of the double mutant.

**Figure 2 ijms-25-09114-f002:**
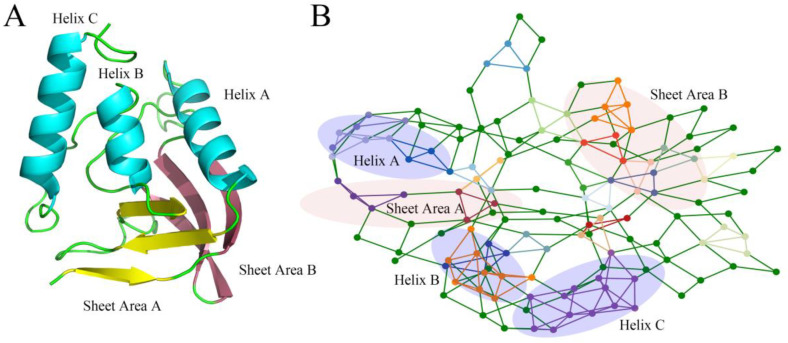
(**A**) The native structure of staphylococcal nuclease in wild-type (WT); (**B**) Distribution of 3-clique community in the residual-contact network of the staphylococcal nuclease. The highlighted elements in the network are the nodes and edges that make up the 3-clique community. Purple-shaded areas are protein helix regions, and light-pink-shaded areas are protein sheets.

**Figure 3 ijms-25-09114-f003:**
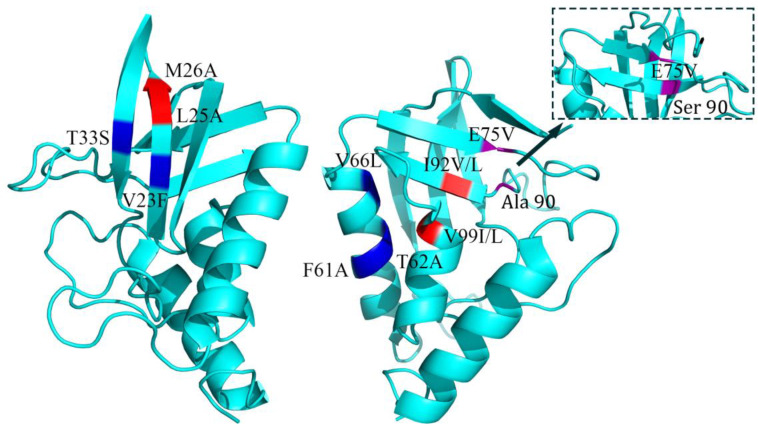
Distribution of double mutation sites in the native staphylococcal nuclease structure that are additive and the 3-clique communities. Ala90 has a lengthened β-sheet after mutation to Ser 90, and in the A90S or E75V/A90S mutant structures, amino acids at positions 75 and 90 are present in the β-sheet structure. Both L25A and M26A are marked in red.

**Figure 4 ijms-25-09114-f004:**
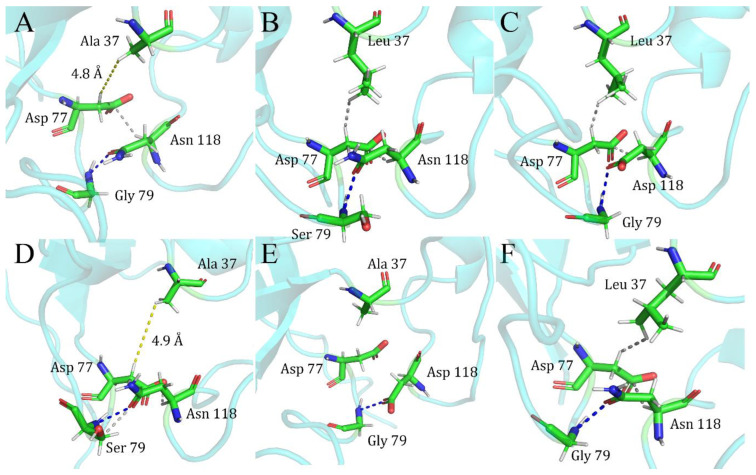
Local contact interactions in single and double mutations of staphylococcal nuclease. (**A**) L37A mutant; (**B**) G79S mutant; (**C**) N118D mutant; (**D**) L37A/G79S double mutant; (**E**) L37A/N118D mutant; (**F**) wild-type. Gray connections represent van der Waals forces, blue are hydrogen bonds, and yellow are distances between atoms in the residue.

**Figure 5 ijms-25-09114-f005:**
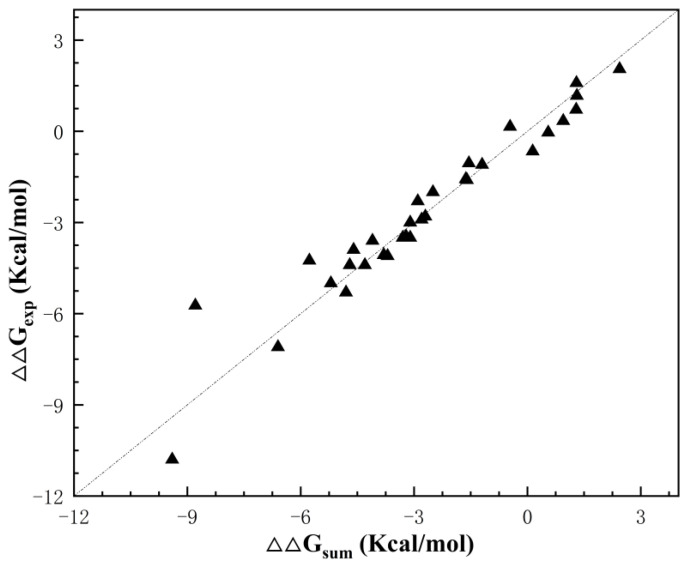
Distribution of thermodynamic data for gene V double mutations.

**Figure 6 ijms-25-09114-f006:**
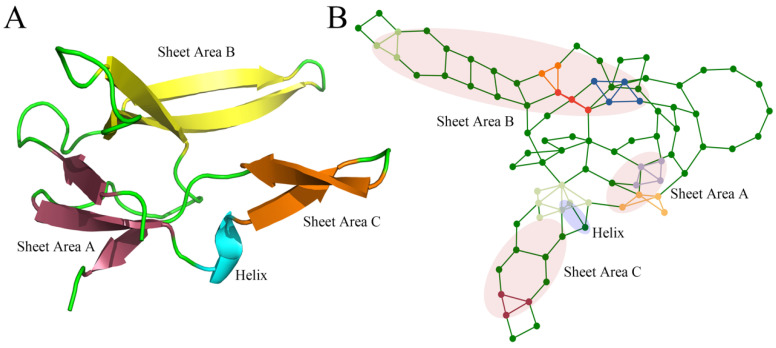
(**A**) The native structure of the gene V protein; (**B**) Distribution of 3-clique community in the residual-contact Network of gene V. The highlighted elements are the nodes and edges that comprise the cliques. Purple-shaded regions are protein helix regions, and light-pink-shaded regions are protein sheet regions.

**Figure 7 ijms-25-09114-f007:**
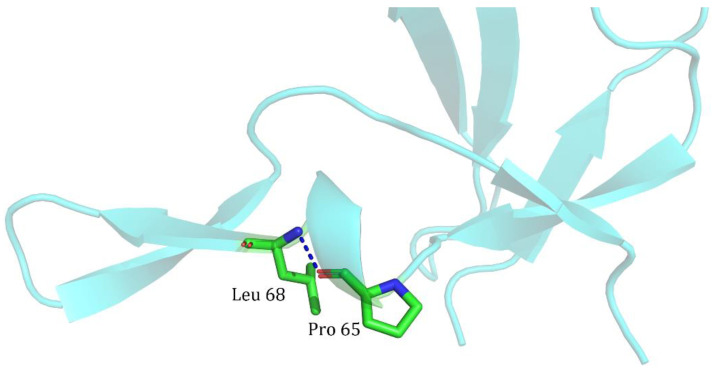
Hydrogen bonding force patterns between the L65P/F68L sites.

**Figure 8 ijms-25-09114-f008:**
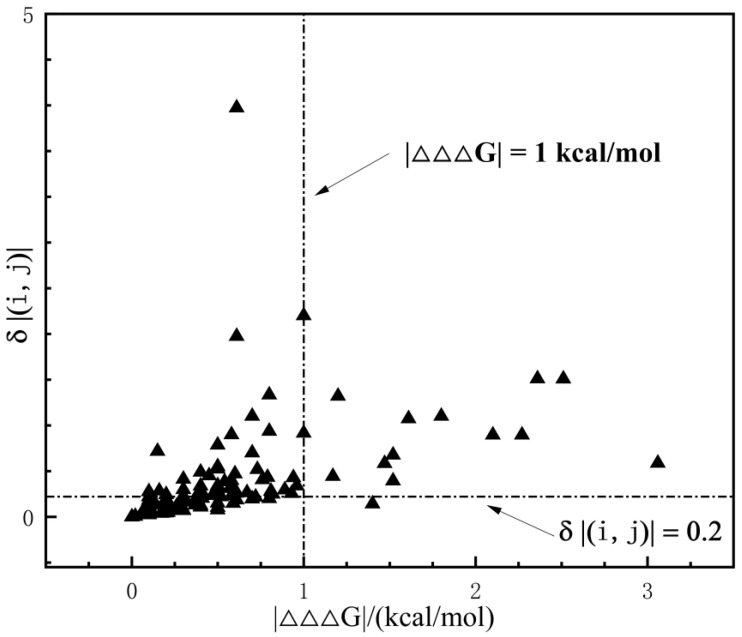
Distribution of double mutant thermodynamic data in staphylococcal nuclease and phage F1 gene V proteins under two ways of judgment.

**Table 1 ijms-25-09114-t001:** Clique communities and additive double mutants of staphylococcal nuclease.

Index	Mutation	∆∆G_sum_	∆∆G_exp_	∆∆∆G	3-Clique Community
1	L37A/A90S	−3.58	−3.32	0.26	**99**, 100, **37**, **36**, 90, **92**
2	T62A/V66L	−2.5	−2.4	0.1	64, **65**, **66**, 67, 68, **69**, **102**, 54, **55**, **56**, 57, **58**, 59, **60**, **61**, **62**, 63
3	I92V/V99I	−0.6	−0.7	−0.1	**98**, **99**, 100, 101, **103**, **104**, 105, **92**, 93
4	I92L/V99I	−0.8	−0.9	−0.1	**98**, **99**, 100, 101, **37**, **103**, **104**, **36**, 105, **102**, 106, **92**
5	I92V/V99L	−0.7	−0.9	−0.2	**98**, **99**, 100, 101, **102**, **103**, **104**, 105, 106, 93, **92**, **61**
6	I92L/V99L	−0.9	−1.3	−0.4	**99**, 100, **36**, **37**, **92**
7	F61A/T62A	−4.8	−4.6	0.2	64, **65**, **66**, 67, **69**, **60**, **61**, **62**, 63
8	L25A/M26A	−4.4	−4.4	0	**25**, **26**, **12**, 13
9	E75V/A90S	−4.21	−3.6	0.61	91, **90**, **75**
10	V23F/T33S	−3.3	−2.72	0.58	33, **34**, **23**

The bold numbers indicate hydrophobic amino acids.

**Table 2 ijms-25-09114-t002:** Clique communities in non-additive double mutants of staphylococcal nuclease.

Index	Mutation	ΔΔG1	ΔΔG2	ΔΔG_sum_	ΔΔG_exp_	ΔΔΔG	3-Clique Community
1	L37A/G79S	−1.68	−2.66	−4.34	−1.83	2.51	\
2	L37A/N118D	−1.68	−2.4	−4.08	−1.72	2.36	\
3	G79S/N118D	−2.66	−2.4	−5.06	−2.79	2.27	79, 80, 118
4	G79D/N118D	−2.28	−2.4	−4.68	−2.58	2.1	79, 80, 118
5	L7A/L37A	−1.58	−1.68	−3.26	−1.65	1.61	\
6	L7A/G79S	−1.58	−2.66	−4.24	−2.77	1.47	\
7	L37A/E75V	−1.68	−2.31	−3.99	−2.47	1.52	\
8	P117L/N118D	0.2	−2.4	−2.2	−1	1.2	77, 78, 79, 117, 118, 119, 120
9	V23F/I72V	−2.3	−1.77	−4.07	−2.9	1.17	\

**Table 3 ijms-25-09114-t003:** Clique communities in phage T4 lysozymes.

Index	Mutation	ΔΔG_sum_	ΔΔG_exp_	ΔΔΔG	3-Clique Community
1	E45A/K48A	−0.55	0.01	0.56	33, 38–50
2	N116A/M120A	−0.03	0.21	0.24	115–125
3	R119A/Q123A	−0.4	−0.17	0.23	84, 111, 114–125
4	M120A/Q122A	−0.44	−0.25	0.19	114–123
5	T115A/R119A	−0.32	−0.17	0.15	114–123
6	E128A/V131A	0.43	0.44	0.01	117, 127–129, 131–133
7	N116D/R119M	0.7	0.6	−0.1	115–116,119–120
8	T115A/S117A	1.13	0.95	−0.18	114–125
9	A98V/T152S	−7.5	−4.8	2.7	2–12, 88–106, 145, 148–155,161
10	M102L/V111F	−2.51	−2.11	0.4	102–103, 106–107, 108,111
11	L99A/E108V	−3.3	−3.1	0.2	92, 95–109, 111–112, 114–125, 137–138, 140–142, 145–146, 153
12	L99G/E108V	−5.6	−5.6	0	1–2, 5–7, 9–11, 92, 95–103, 105–109, 111–112, 114, 138, 145–146, 148–150, 153–154
13	M6I/R96H	−4.2	−5.08	−0.88	1–3, 5–7, 9–11, 85–93, 95–106, 130, 144–145, 148–150, 152–154

## Data Availability

Data are contained within the article and Appendix A.

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
