# Peer review of "Analysis of Correlation Effects of Double Mutations in Enzymes: A Revised Residual-Contact Network Clique Model"

_ijms, 2024, doi:10.3390/ijms25169114_

Round 1

Reviewer 1 Report

Comments and Suggestions for Authors

Major points:

Q1. In the introduction, it should give the backgrounds on the staphylococcal nuclease.

Q2. Why does it select the staphylococcal nuclease from S. aureus and the 86 Gene V protein from phage f1 for this study?

Q3. Which tool is used to prepare Figure 2 and Figure 6?

Q4. Figure 3 should be improved. It should display the amino acid of the marked sites.

Q5. Please pay much attention to the organization and writing of the text. There are too many error text Error! Reference source not found. See, lines 111, 119, 212, 213,218, 219, 257, etc.

Q6. Why do you not perform the MD experiment? I may suggest the authors give the RMSF and RMSD data.

Q7. The conditions in section “METHODs” are not clear.

Q8. To improve the attractiveness of reading, the organization and the text of results and discussion should be improved carefully. Also, conclusions should be improved carefully.

Q9. The writing of this submission should be improved carefully.  Please also correct the mistakes throughout this text. The English of this manuscript must be improved before publication. Your manuscript also requires revision concerning the language used. It is suggested that you obtain assistance from a colleague who is well-versed in English or whose native language is English.

Comments on the Quality of English Language

The writing of this submission should be improved carefully.  Please also correct the mistakes throughout this text. The English of this manuscript must be improved before publication. Your manuscript also requires revision concerning the language used. It is suggested that you obtain assistance from a colleague who is well-versed in English or whose native language is English.

Reviewer 2 Report

Comments and Suggestions for Authors

The authors provided the revised residual-contact network clique model for understanding enzyme double mutation additivity. The study has direct applications in enzyme design and optimization, which are critical for bio-industrial applications. The availability of the code on GitHub promotes transparency and allows for further research and validation by others. However, the manuscript has some limitations. While the model successfully identifies a high percentage of additive and non-additive mutations, the authors do not thoroughly address the limitations or potential biases in the model. In addition, the studies rely heavily on computational predictions. Although the predictions are compared with existing experimental data, additional experimental validation would strengthen the conclusions. To validate Alphafold models, structure alignments with the wile type structure are helpful. After addressing the issues mentioned, I recommend proceeding with the publication.

Round 2

Reviewer 1 Report

Comments and Suggestions for Authors

No further issues.

Reviewer 2 Report

Comments and Suggestions for Authors

The manuscript adequately addresses all relevant issues. Therefore, I recommend the manuscript for publication.